# Genome-wide association study in quinoa reveals selection pattern typical for crops with a short breeding history

Dilan SR Patiranage[1], Elodie Rey[2], Nazgol Emrani[1]*, Gordon Wellman[2], Karl Schmid[3], Sandra M Schmöckel[4], Mark Tester[2], Christian Jung[1]*

[1]Plant Breeding Institute, Christian-Albrechts-University of Kiel, Kiel, Germany; [2]King Abdullah University of Science and Technology (KAUST), Biological and Environmental Sciences & Engineering Division (BESE), Thuwal, Saudi Arabia; [3]Institute of Plant Breeding, Seed Science and Population Genetics, University of Hohenheim, Stuttgart, Germany; [4]Department of Physiology of Yield Stability, University of Hohenheim, Stuttgart, Germany

*For correspondence:
n.emrani@plantbreeding.uni-kiel.de (NE);
c.jung@plantbreeding.uni-kiel.de (CJ)

**Competing interest:** The authors declare that no competing interests exist.

**Abstract** Quinoa germplasm preserves useful and substantial genetic variation, yet it remains untapped due to a lack of implementation of modern breeding tools. We have integrated field and sequence data to characterize a large diversity panel of quinoa. Whole-genome sequencing of 310 accessions revealed 2.9 million polymorphic high confidence single nucleotide polymorphism (SNP) loci. Highland and Lowland quinoa were clustered into two main groups, with $F$ST divergence of 0.36 and linkage disequilibrium (LD) decay of 6.5 and 49.8 kb, respectively. A genome-wide association study using multi-year phenotyping trials uncovered 600 SNPs stably associated with 17 traits. Two candidate genes are associated with thousand seed weight, and a resistance gene analog is associated with downy mildew resistance. We also identified pleiotropically acting loci for four agronomic traits important for adaptation. This work demonstrates the use of re-sequencing data of an orphan crop, which is partially domesticated to rapidly identify marker-trait association and provides the underpinning elements for genomics-enabled quinoa breeding.

## Editor's evaluation

This is a comprehensive study of genomic and phenotypic diversity in the orphan crop quinoa. Based on whole genome resequencing of 310 accessions and field phenotyping of the same set of accessions for two years, the study identified the genetic basis of agronomically important traits. Based on this promising work, there will likely be scope for quick improvement of this orphan crop through breeding.

## Introduction

Climate change poses a great threat to crop production worldwide. In temperate climates of the world, higher temperatures and extended drought periods are expected. Moreover, crop production in industrialized countries depends on only a few major crops resulting in narrow crop rotations. Therefore, rapid transfer of wild species into crops using genetic modification and targeted mutagenesis is currently discussed (*Li et al., 2018*; *Stetter et al., 2017*). Alternatively, orphan crops with a long tradition of cultivation but low breeding intensity can be genetically improved by genomics-assisted selection methods. Quinoa (*Chenopodium quinoa* Willd.) is a pseudocereal crop species with a long history of cultivation. It was first domesticated about 5000–7000 years ago in the Andean

**eLife digest** As human populations grow and climate change tightens its grip, developing nutritious crops which can thrive on poor soil and under difficult conditions will become a priority. Quinoa, a harvest currently overlooked by agricultural research, could be an interesting candidate in this effort.

With its high nutritional value and its ability to tolerate drought, frost and high concentrations of salt in the soil, this hardy crop has been cultivated in the Andes for the last 5,000 to 7,000 years. Today its commercial production is mainly limited to Peru, Bolivia, and Ecuador. Pinpointing the genetic regions that control traits such as yields or flowering time would help agronomists to create new varieties better suited to life under northern latitudes and mechanical farming.

To identify these genes, Patiranage et al. grew 310 varieties of quinoa from all over the world under the same conditions; the genomes of these plants were also examined in great detail. Analyses were then performed to link specific genetic variations with traits relevant to agriculture, helping to pinpoint changes in the genetic code linked to differences in how the plants grew, resisted disease, or produced seeds of varying quality. Candidate genes likely to control these traits were then put forward.

The study by Patiranage et al. provides a genetic map where genes of agronomical importance have been precisely located and their effects measured. This resource will help to select genetic profiles which could be used to create new quinoa breeds better adapted to a changing world.

region. Quinoa was a staple food during the pre-Columbian era, and the cultivation declined after the introduction of crops like wheat and barley by the Spanish rulers. Owing to diversity, biotic and abiotic stress tolerance, and ecological plasticity, quinoa can adapt to a broad range of agroecological regions (*González et al., 2015*; *Ruiz et al., 2013*). Due to its high seed protein content and favorable amino acid composition, the nutritional value is even higher than beef, fish, and other major cereals (*Abugoch James, 2009*; *Vega-Gálvez et al., 2010*). These favorable characteristics contributed to the increasing worldwide popularity of quinoa among consumers and farmers.

A spontaneous hybridization event between two diploid species between 3.3 and 6.3 million years ago gave rise to the allotetraploid species quinoa ($2n = 4x = 36$) with a genome size of 1.45–1.5 Gb (nuclear DNA content 1C=1.49 pg) (*Kolano et al., 2011*; *Palomino et al., 2008*). A reference genome of the coastal Chilean quinoa accession PI 614886 has been published with 44,776 predicted gene models and whole-genome re-sequencing of *Chenopodium pallidicaule* and *Chenopodium suecicum* species, close relatives of the A and B subgenome donor species, respectively (*Jarvis et al., 2017*). The organellar genomes are originated from the A-genome ancestor (*Maughan et al., 2019*).

Quinoa belongs to the Amaranthaceae, together with some other economically important crops like sugar beet, red beet, spinach, and amaranth. It reproduces sexually after self-pollination. Quinoa accessions are typically homozygous inbred lines. Nonetheless, heterozygosity in some accessions has been reported, indicating cross-pollination (*Christensen et al., 2007*). The inflorescences are panicles, which are often highly branched. Florets are tiny, which is a significant obstacle for hand-crossing. However, routine protocols for $F_1$ seed production in combination with marker-assisted selection have been developed recently (*Emrani et al., 2020*; *Peterson et al., 2015*).

Systematic breeding of quinoa is still in its infancy compared to major crops. Until recently, breeding has been mainly limited to Bolivia (*Gandarillas, 1979*) and Peru (*Bonifacio et al., 2013*), the major growing quinoa areas. Therefore, quinoa can be regarded as a partially domesticated crop. Many accessions suffer from seed shattering, branching, and non-appropriate plant height (PH), typical domestication traits. The major breeding objectives are apart from these characters: grain yield and seed size, downy mildew resistance, synchronized maturity, stalk strength, and low saponin content (*Gomez-Pando, 2015*). In the past years, activities have been intensified to breed quinoa genotypes adapted to temperate environments, for example, Europe, North America, and China (*Murphy, 2018*). Here, the major problem is adapting to long-day conditions because quinoa is predominantly a short-day plant due to its origin from regions near the equator.

There are only a few studies about the genetic diversity of quinoa. They were mainly based on phenotypic observations (*Gandarillas et al., 1979*; *Wilson, 1988*) and low throughput marker systems like random amplified polymorphic DNA (*Ruas et al., 1999*), amplification fragment length

polymorphisms (*Rodríguez and Isla, 2009*), and microsatellites (*Mason et al., 2005*). *Maughan et al., 2012*, used five bi-parental populations to identify ca. 14,000 single nucleotide polymorphisms (SNPs), from which 511 KASP markers were developed. Genotyping 119 quinoa accessions gave the first insight into the population structure of this species (*Maughan et al., 2012*). Now, the availability of a reference genome enables genome-wide genotyping. *Jarvis et al., 2017*, re-sequenced 15 accessions and identified ca. 7.8 million SNPs. In another study, 11 quinoa accessions were re-sequenced, and 8 million SNPs and ca. 842,000 indels were identified (*Zhang et al., 2017*).

Our study aimed to analyze the population structure of quinoa and patterns of variation by re-sequencing a diversity panel encompassing germplasm from all over the world. Using millions of markers, we performed a genome-wide association study using multiple year field data. Here, we identified QTLs (quantitative trait loci) that control agronomically important traits important for breeding cultivars to be grown under long-day conditions. Our results provide information for further understanding the genetic basis of agronomically important traits in quinoa and will be instrumental for future breeding.

## Results

### Re-sequencing 310 quinoa accessions reveal high sequence variation

We assembled a diversity panel made of 310 quinoa accessions representing regions of major geographical distributions of quinoa (*Figure 1—figure supplement 1*). The diversity panel comprises accessions with different breeding histories (*Supplementary file 1a*). We included 14 accessions from a previous study, of which 7 are wild relatives (*Jarvis et al., 2017*). The mean mapped read depth ranged from 4.07 to 14.55, with an average of 7.78, indicating an adequate mapping quality required for accurate SNP calling despite the relatively modest sequencing depth. We mapped sequence reads to the reference genome V2 (CoGe id60716). Using mapping reads, we identified 45,330,710 unfiltered SNPs.

After filtering the initial set of SNPs, we identified 4.5 million SNPs in total for the base SNP set. We further filtered the SNPs for MAF >5% (HCSNPs). We obtained 2.9 million high confidence SNPs for subsequent analysis (*Supplementary file 1b*). Across the whole genome, the average SNP density was 2.39 SNPs/kb. However, SNP densities were highly variable between genomic regions and ranged from 0 to 122 SNPs/kb (*Figure 1—figure supplement 2*). We did not observe significant differences in SNP density between the two subgenomes (a subgenome 2.43 SNPs/kb; B subgenome 2.35 SNPs/kb). Moreover, we did not see any correlation between sequencing depth and heterozygosity (*Figure 1—figure supplement 2b*), which indicates an adequate mapping quality required for accurate SNP calling. In an additional analysis, we divided the filtered SNPs into homozygous and heterozygous SNPs for each sample. Then, we calculated the mean read depth (DP) and genotype quality (GQ) of each sample separately for the homozygous and heterozygous fraction of the genome (*Figure 1—figure supplement 3*). Mean GQ of the heterozygous SNP calls was 61.34, whereas the

**Table 1.** Summary statistics of genome-wide single nucleotide polymorphisms identified in 303 quinoa accessions.

| Parameter | Type | All genotypes (quinoa only) | Highland population | Lowland population |
|---|---|---|---|---|
| SNP | Total | 2,872,935 | 2,590,907 | 1,938,225 |
| | Population-specific SNPs | | 1,512,301 | 859,619 |
| | Intergenic | 2,452,347 | 2,227,952 | 1,649,310 |
| | Introns | 251,481 | 101,546 | 172,692 |
| | Exons | 114,654 | 214,945 | 78,248 |
| Nucleotide diversity | | | $5.78 \times 10^{-4}$ | $3.56 \times 10^{-4}$ |
| Tajima's *D* | | | 0.884 | –0.384 |
| Population divergences | $F_{ST}$ (weighted average) | | 0.36 | |

mean GQ of homozygous SNP calls was 21.19, indicating that a higher stringency was used for the heterozygous SNP calls. We also compared the DP with the GQ for both filtered and unfiltered SNPs. The results indicated that higher GQ values were used for low DP regions in order to ensure correct genotype calls. Then, we split the SNPs by their functional effects as determined by SnpEff (*Cingolani et al., 2012a*). Among SNPs located in non-coding regions, 598,383 and 617,699 SNPs were located upstream (within 5 kb from the transcript start site) and downstream (within 5 kb from the stop site) of a gene, whereas 114,654 and 251,481 SNPs were located within exon and intron sequences, respectively (*Table 1*). We further searched for SNPs within coding regions. We found 70,604 missense SNPs and 41,914 synonymous SNPs within coding regions of 53,042 predicted gene models.

## Linkage disequilibrium and population structure of the quinoa diversity panel

Across the whole genome, linkage disequilibrium (LD) decay between SNPs averaged 32.4 kb (at $r^2$=0.2). We did not observe substantial LD differences between subgenome A ($r^2$=0.2 at 31.9 kb) and subgenome B ($r^2$=0.2 at 30.7 kb) (*Figure 1—figure supplement 4*). The magnitude of LD decay among chromosomes did not vary drastically except for chromosome Cq6B, which exhibited a substantially slower LD decay (*Figure 1—figure supplement 4a b*).

Then, we unraveled the population structure of the diversity panel. We performed principal component (PCA$_{(SNP)}$), population structure, and phylogenetic analyses. PCA$_{(SNP)}$ showed two main clusters consistent with previous studies (*Christensen et al., 2007*). The first and second principal components (PC1$_{(SNP)}$ and PC2$_{(SNP)}$) explained 23.35% and 9.45% of the variation, respectively (*Figure 1a*); 202 (66.67%) accessions were assigned to subpopulation 1 (SP1) and 101 (33.33%) to subpopulation 2 (SP2). SP1 comprised mostly Highland accessions, whereas Lowland accessions were found in SP2. PCA demonstrated a higher genetic diversity of the Highland population (*Figure 1a*). We also calculated PCs for each chromosome separately. For 16 chromosomes, the same clustering as for the whole genome was calculated. Nevertheless, two chromosomes, Cq6B and Cq8B, showed three distinct clusters (*Figure 1—figure supplement 5*). This is due to the split of the Lowland population into two clusters. We reasoned that gene introgressions on these two chromosomes from another interfertile group might have caused these differences. This is also supported by a slower LD decay on chromosome Cq6B (*Figure 1—figure supplement 4b*). This discrepancy also might arise due to the Lowland reference genome used for mapping the reads in this study (CoGe id60716), which may have structural differences compared to the genomes of Highland accessions.

We also performed a population structure analysis with the ADMIXTURE software. We used 10 independent sets of 50,000 randomly chosen SNPs. Then, we performed ADMIXTURE analysis for each subset separately with a predefined number of genetic clusters *K* from 2 to 10 and different random seeds with 1000 bootstraps. The Q-matrices obtained were aligned with the greedy algorithm in the CLUMPP software (*Jakobsson and Rosenberg, 2007*). We used cross-validation to estimate the most suitable number of populations. Cross-validation error decreased as the *K* value increased, and we observed that after *K*=5, cross-validation error reached a plateau (*Figure 1—figure supplement 6b*). We observed allelic admixtures in some accessions, likely owing to their breeding history. The wild accessions were also clearly separated at the smallest cross-validation error of *K*=8, except two *Chenopodium hircinum* accessions (*Figure 1c*). This could be because *C. hircinum* is the closest crop wild relative; it also may have outcrossed with quinoa. The Highland population was structured into five groups, while the Lowland accessions were split into two subpopulations. The broad agro-climatic diversity of the Andean Highland germplasm might have caused a higher number of subpopulations.

For clustering accessions based on sequence polymorphism, we combined 10 subsets created for ADMIXTURE analysis and removed redundant SNPs among subsets. We analyzed the relationships between quinoa accessions using 434,077 SNPs. Constructing a maximum likelihood (ML) tree gave rise to five clades (*Figure 2*). We found that the placement of the wild quinoa species as distant outgroups was concordant with the previous reports confirming that quinoa was domesticated from *C. hircinum* (*Jarvis et al., 2017*). However, we found that *the C. hircinum* accession BYU 566 (from Chile) was placed at the base of both Lowland and Highland clades, which contrasts to *Jarvis et al., 2017*, where this accession was placed at the base of Lowland (coastal) quinoa. As expected, accessions from the USA and Chile are closely related because the USDA germplasm had been collected at these geographical regions.

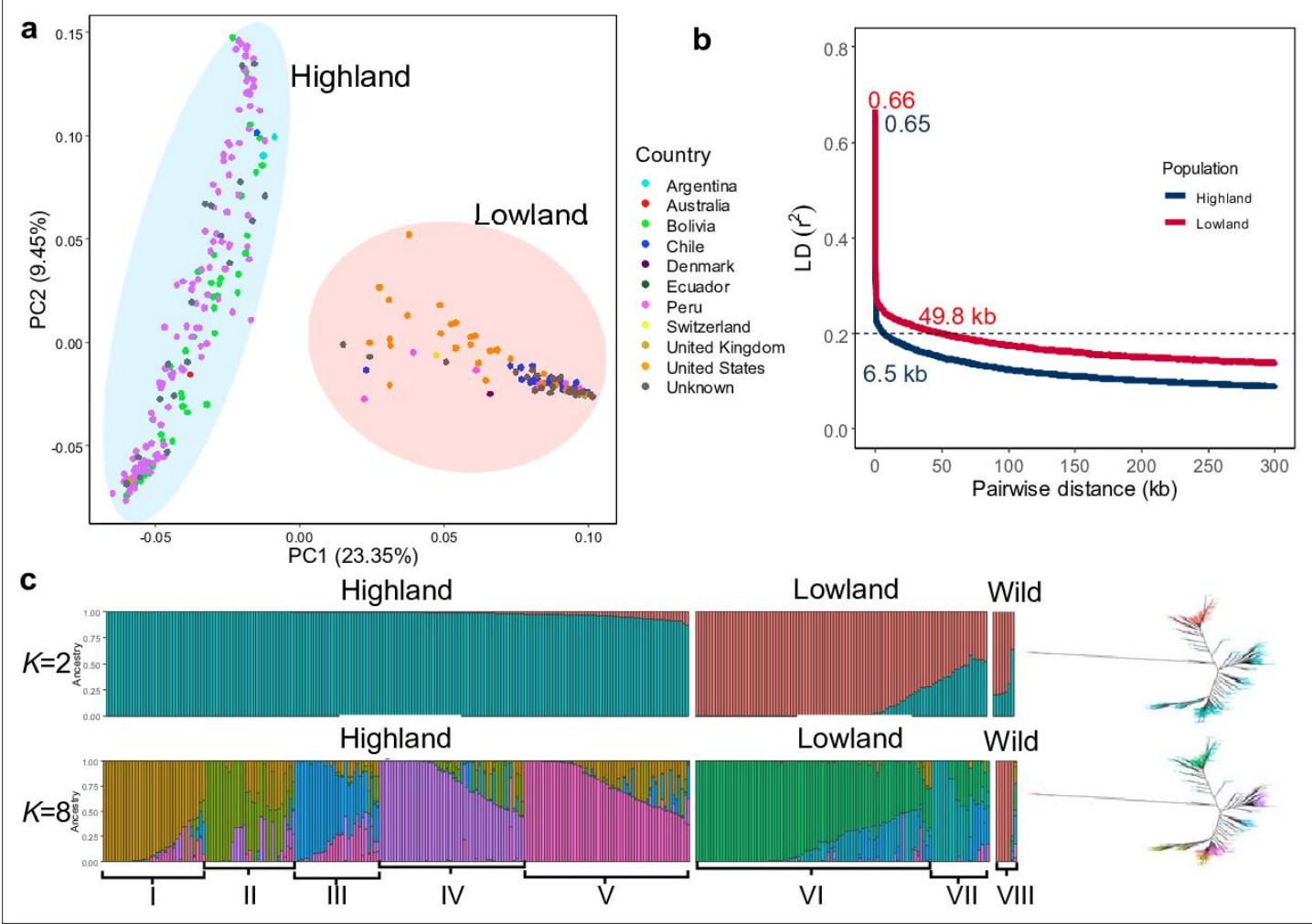

**Figure 1.** Genetic diversity and population structure of the quinoa diversity panel. (**a**) Principal component analysis (PCA) of 303 quinoa accessions. PC1 and PC2 represent the first two analysis components, accounting for 23.35% and 9.45% of the total variation, respectively. The colors of dots represent the origin of accessions. Two populations are highlighted by different colors: Highland (light blue) and Lowland (pink). (**b**) Subpopulation-wise linkage disequilibrium (LD) decay in Highland (blue) and Lowland population (red). (**c**) Population structure is based on 10 subsets of SNPs, each containing 50,000 single nucleotide polymorphisms (SNPs) from the whole-genome SNP data. Model-based clustering was done in ADMIXTURE with different numbers of ancestral kinships (*K*=2 and *K*=8). *K*=8 was identified as the optimum number of populations. Left: Each vertical bar represents an accession, and color proportions on the bar correspond to the genetic ancestry. Right: Unrooted phylogenetic tree of the diversity panel. Colors correspond to the subpopulation.

The online version of this article includes the following source data and figure supplement(s) for figure 1:

**Source data 1.** Principal component analysis (PCA) of 303 quinoa accessions.

**Source data 2.** Subpopulation-wise linkage disequilibrium (LD) decay in Highland and Lowland population.

**Source data 3.** Population structure is based on 10 subsets of single nucleotide polymorphisms (SNPs), each containing 50,000 SNPs from the whole-genome SNP data.

**Figure supplement 1.** Geographical origin of the accessions forming the quinoa diversity panel.

**Figure supplement 2.** The SNP distribution in quinoa genome.

**Figure supplement 3.** The effect of read depth on genotype quality for homozygous and heterozygous SNPs.

**Figure supplement 4.** Chromosome-wide linkage disequilibrium (LD) decay in genome A (**a**) and genome B (**b**).

**Figure supplement 5.** Single nucleotide polymorphism (SNP)-based principal component analysis (PCA) across all 18 quinoa chromosomes.

**Figure supplement 6.** population structure analysis in quinoa.

**Figure supplement 7.** Diversity of populations along chromosomes measured based on 10 kb non-overlapping windows.

**Figure supplement 8.** Distribution of Tajima's *D* along chromosomes in Lowland (**a**) and Highland (**b**) populations.

**Figure supplement 9.** Local principal component analysis (PCA) and identification of candidate genes using diversity parameters.

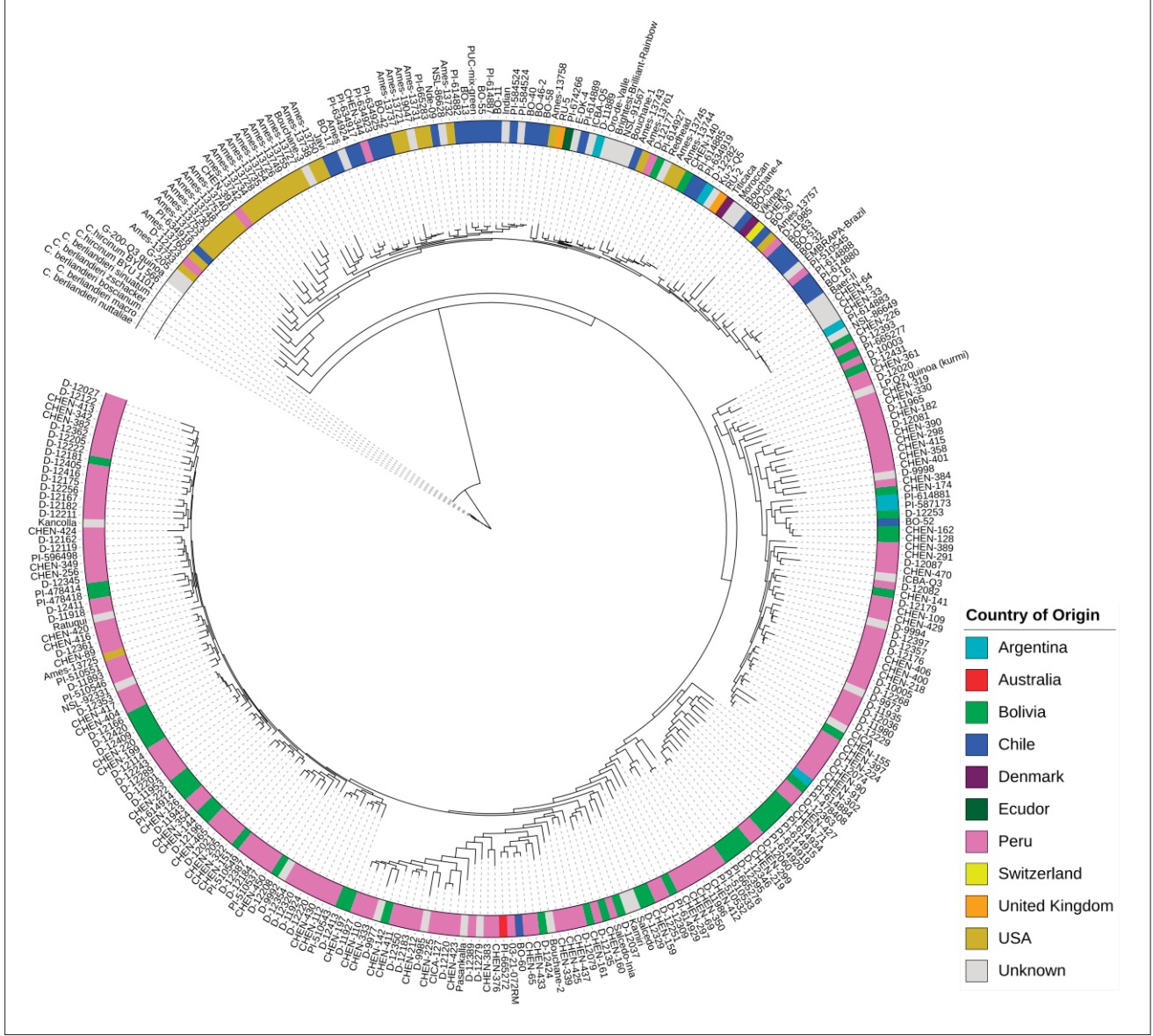

**Figure 2.** Maximum likelihood tree of 303 quinoa and 7 wild *Chenopodium* accessions from the diversity panel. Colors depict the geographical origin of accessions.

The online version of this article includes the following source data and figure supplement(s) for figure 2:

**Source data 1.** Maximum likelihood tree of 303 quinoa and 7 wild *Chenopodium* accessions from the diversity panel.

**Figure supplement 1.** Genetic relationships between quinoa accessions.

## Genomic patterns of variations between Highland and Lowland quinoa

We were interested in patterns of variation in response to geographical diversification. We used PCA derived clusters and phylogenetic analysis to define two diverged quinoa populations (namely Highland and Lowland). These divergent groups are highly correlated with Highland and Lowland geographical origin. We used the base SNP set to analyze diversity statistics. To detect genomic regions affected by the population differentiation, we measured the level of nucleotide diversity using 10 kb non-overlapping windows (*Varshney et al., 2017b*; *Figure 1—figure supplement 7*). Then, we calculated the whole genome-wide LD decay across the two populations (Highland vs. Lowland); LD decayed more rapidly in Highland quinoa (6.5 kb vs. 49.8 kb, at $r^2$=0.2) (*Figure 1b*). To measure nucleotide diversity, we scanned the quinoa genome with non-overlapping windows of 10 kb in length in both populations separately. The nucleotide diversity of the Highland population

(5.78 × 10⁻⁴) was 1.62-fold higher compared to the Lowland population (3.56 × 10⁻⁴) (*Table 1* and *Figure 1—figure supplement 7*). We observed left-skewed distribution and negative Tajima's *D* value (–0.3883) in the Lowland populations indicating recent population growth (*Table 1* and *Figure 1—figure supplement 8*). Genomic regions involved in adaptation to the Highlands should have much lower diversity in the Highland population than in the Lowland population, and genomic regions involved in adaptation to the Lowlands should have lower diversity in the Lowland population compared to the Highland population. Therefore, we calculated the nucleotide diversity ratios between Highland and Lowland to identify major genomic regions underlying the population differentiation (*Figure 1—figure supplement 7*). The $F_{ST}$ value between populations was estimated to be 0.36, illustrating strong population differentiation. Concerning the regions of variants, exonic SNPs are substantially higher in the Highland population (*Table 1* and *Figure 1—figure supplement 7*).

We conducted a local PCA to identify genomic regions with a strong population structure. The genome was divided into 50 kb non-overlapping windows, and PCA was calculated for each window using the lostruct program (*Li and Ralph, 2019*), which calculates a similarity score by comparing PCs obtained from each window. Similarity scores were then stored as a matrix and visualized using multidimensional scaling (MDS) transformation. Strong indications of the population structure are represented in the corners of the MDS analysis; usually, it follows a triangle, providing three corners (corner 1, corner 2, and corner 3) (*Figure 1—figure supplement 9a*). Candidate genomic regions were defined as the 1% of the MDS coordinates closest to each of the corners. They consist of the windows with the strongest genetic differentiation across the genome (*Figure 1—figure supplement 9b*). Then, we selected candidate genomic regions from each corner and calculated the PCs using SNPs present in those regions. SNPs from the candidate genomic regions of corner 1 structured the diversity panel into two clusters (*Figure 1—figure supplement 9c*). Corner 2 also resulted in two clusters, but clustering was not as strong as corner 1 regions (*Figure 1—figure supplement 9d*). Corner 3 separated accessions into three clusters similar to the PCA using the candidate genomic regions obtained from the nucleotide diversity ratio analysis (*Figure 1—figure supplement 9f* and g).

Then, we located the genes within candidate regions obtained from all three analyses. We identified 936, 953, and 546 candidate genes located within critical regions from the nucleotide diversity ratio ($π$ (Highland/Lowland)), $F_{ST}$, and local PCA corner 1 (*Figure 1—figure supplement 9h*). Of these, only four genes were shared among all analyses, and 30 genes were shared between $F_{ST}$ and genomic regions in corner 1 of the local PCA plot. Genomic regions in corner 3 of the local PCA plot and with a high nucleotide diversity ratio shared 102 genes (*Figure 1—figure supplement 9i*).

## Mapping agronomically important trait loci in the quinoa genome

We evaluated 13 qualitative and 4 dichotomous traits on 350 accessions across 2 different years. At the time of the final harvest, 254 accessions did not yet reach maturity (senescence). However, all accessions produced seeds, therefore they could be investigated for seed-related traits. For all traits, substantial phenotypic variation among accessions was found. High heritabilities were calculated for all quantitative traits except for the number of branches (NoB) and stem lying (STL), which indicates that the phenotypic variation between the accessions is caused mainly by genetic variation (*Supplementary file 1c*). Trait correlations between years were also high (*Figure 3—figure supplement 2*), which is following the heritability estimates. We found the strongest positive correlation between days to maturity (DTM) and panicle length (PL), and PH, whereas the strongest negative correlation was found between DTM and thousand seed weight (TSW) (*Figure 3—figure supplement 3*). Then, a PCA was performed based on 12 quantitative traits (PCA(PHEN)) to explore the phenotypic relationship among quinoa accessions. The first two PCs explained 62.12% of the phenotypic variation between the accessions. The score plot of the PCs showed a similar clustering pattern as the SNP-based PCA (PCA(SNP)) (*Figure 1* and *Figure 3—figure supplement 4a*). PCA(PHEN) variables factor map indicated that most Lowland accessions were high yielding with high TSW and dense panicles. Moreover, these accessions are early flowering and early maturing, and they are short (*Figure 3—figure supplement 4b*). Phenotype-based PCA(PHEN) also showed that the Lowland accessions are better adapted/selected for cultivation in long-day photoperiods than the Highland accessions. These results are in accordance with LD, nucleotide diversity, and Tajima's *D* estimations, implying the Lowland accessions underwent a stronger selection during breeding.

Then, we calculated the best linear unbiased estimates (BLUE) of the traits investigated. In total, 294 accessions shared the re-sequencing information and phenotypes out of 350 phenotypically evaluated accessions. For GWAS analysis, we used ~2.9 million high confidence SNPs. We considered pairwise kinship value distribution to determine that all accessions could be used for GWAS analysis without conducting subpopulation-wise analyses (*Figure 3—source data 4*). In total, we identified 1480 significant (suggestive threshold: p<9.41e-7) SNP-trait associations (marker-trait association [MTAs]) for 17 traits (*Figure 3—source data 4*). The number of MTAs ranged from 4 (STL) to 674 (DTM) (*Supplementary file 1d*). In agreement with previous reports, we defined an MTA as 'consistent' when detected in both years (*Varshney et al., 2019*). We identified 600 consistent MTAs across 11 traits. TSW and DTM showed the highest number of 'consistent' associations. Among these, 143 MTAs are located within a gene, and 22 SNPs resulted in a missense mutation (*Supplementary file 1e*). MTAs for the duration from bolting to flowering (days to bolting to days to flowering), number of branching, seed yield, STL, and growth type (GT) were not 'consistent' between years (*Figure 3—source data 4*). This is also reflected by the low estimates of heritability for these traits, indicating considerably higher genotype × environment interactions. Using the SNPs not located in the repetitive regions, we identified 619 MTA across 11 traits, of which 291 MTAs are common between both analyses (*Figure 3—figure supplement 7*). Unique associations of 476 and 328 were identified from whole-genome SNPs and repeat masked SNPs, respectively. However, the comparison of GWAS results of PCs between whole-genome SNP set and repeat masked SNP set showed that highly significant associations could be identified even if the repetitive regions were excluded from the analysis (*Figure 3—figure supplement 7b and c*).

## Candidate genes for agronomically important traits

First, we tested the resolution of our mapping study. We searched for candidate genes 50 kb down- and upstream of significant SNPs for two qualitative traits in quinoa, flower color and seed saponin content. To define candidate genes, we considered homologous genes that have already been functionally characterized in other species. We identified highly significant MTAs for stem color on chromosome Cq1B (69.72–69.76 Mb). Two genes (*CqCYP76AD1* and *CqDODA1*) in this region exhibit high sequence homology to betalain synthesis pathway genes *BvCYP76AD1* (*Hatlestad et al., 2012*) and *BvDODA1* (*Bean et al., 2018*) from sugar beet (*Figure 4—figure supplement 1*). A significant MTA for saponin content on chromosome Cq5B between 8.85 and 9.2 Mb included 29 genes, of which the two *BHLH25* were in LD with the significantly associated SNPs. *BHLH25* genes were reported to control saponin content in quinoa (*Jarvis et al., 2017*; *Figure 4—figure supplement 1b*). This demonstrates that the marker density is high enough to narrow down to causative genes underlying a trait.

Then, we examined four quantitative traits. We obtained seven MTAs on chromosome Cq2A with the traits days to flowering, DTM, PH, and PL indicating pleiotropic gene action (*Figure 3a* and *Supplementary file 1f*). To further investigate genes that are pleiotropically active on different traits, we followed a multivariate approach (*Solovieff et al., 2013*). First, we performed a PCA of the four phenotypes (cross-phenotypes; genetically correlated traits). We found 89.94% of the variation could be explained by the first two PCs of the cross-phenotypes (PCA$_{(CP)}$) (*Figure 3—figure supplement 5*), which suggests that PCA $_{(CP)}$ is suitable to reduce dimensions for a GWAS of cross-phenotypes. Since the PCA$_{(CP)}$ revealed a similar clustering as PCA$_{(SNP)}$, these analyses results provide preliminary indications that in quinoa, days to flowering, DTM, PH, and PL are highly associated with population structure and may reflect adaptation to diverse environments. Then, we performed a GWAS analysis using the first three PCs as traits (PC-GWAS) (*Figure 3—figure supplement 5c*). We identified strong associations on chromosomes Cq2A, Cq7B (PC1), and Cq8B (PC2) (*Figure 3—figure supplement 6*). Of 468 MTAs (PC1: 426 and PC2: 42) across the whole genome, 222 (PC1: 211 and PC2: 11) are located within 95 annotated genes. We found 14 SNPs that changed the amino acid sequence in 12 predicted protein sequences of associated genes (*Supplementary file 1e*). In the next step, we searched genes located 50 kb flanking to an MTA, considering a threshold that is below the genome-wide LD of the Lowland population. Altogether, 605 genes were identified (PC1: 520 and PC2: 85) (*Supplementary file 1g*).

We found the region 8.05–8.15 Mb on chromosome Cq2A to be of special interest because it displays stable pleiotropic MTAs for days to flowering, DTM, PH, and PL. We identified five genes

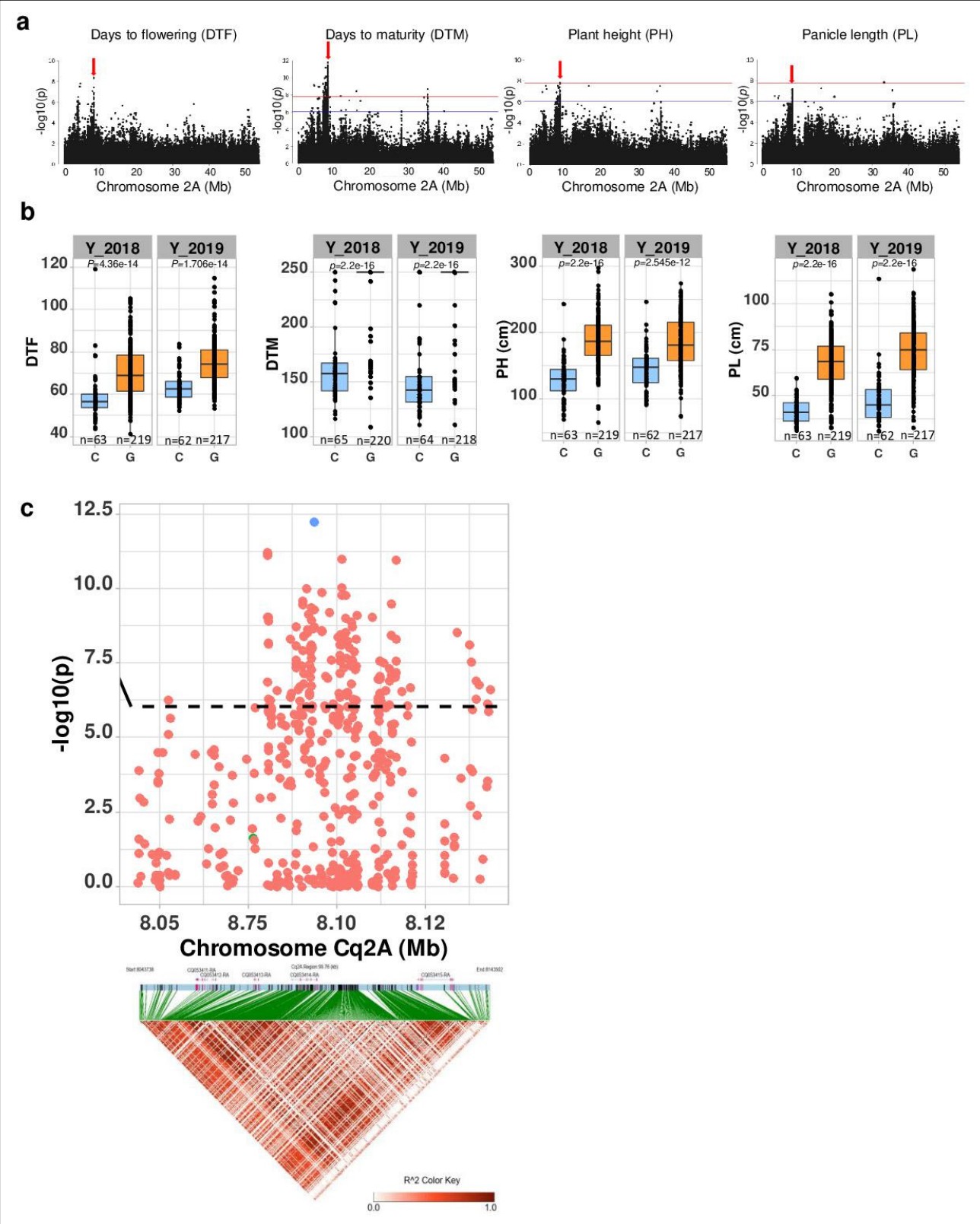

**Figure 3.** Genomic regions associated with important agronomic traits. (**a**) Significant marker-trait associations (MTAs) for days to flowering (DTF), days to maturity (DTM), plant height (PH), and panicle density on chromosome Cq2A. Red color arrows indicate the single nucleotide polymorphism (SNP) loci pleiotropically acting on all four traits. (**b**) Boxplots showing the average performance for four traits over 2 years, depending on single nucleotide variation (C or G allele) within locus Cq2A_ 8093547. The P-values written above the boxplot are from Wilcoxon mean comparisons test (unpaired)

*Figure 3 continued on next page*

*Figure 3 continued*

between C and G allele. (**c**) Local Manhattan plot from region 8.04–8.14 Mb on chromosome Cq2A associated with PC1 of the DTF, DTM, PH, and panicle length (PL), and local linkage disequilibrium (LD) heatmap (bottom). The triangle below is the LD heatmap and the colors represent the pairwise correlation ($r^2$) between individual SNPs. On top of the triangle, gene models are represented. Green color dots represent the strongest MTA (Cq2A_8093547).

The online version of this article includes the following source data and figure supplement(s) for figure 3:

**Source data 1.** Genomic regions associated with important agronomic traits.

**Source data 2.** Boxplots showing the average performance for four traits over 2 years.

**Source data 3.** Local Manhattan plot from region 8.04–8.14 Mb on chromosome Cq2A associated with PC1 of the days to flowering, days to maturity, plant height, and panicle length.

**Source data 4.** Manhattan plots from GWAS with data from 2018 (left), 2019 (center), and the mean of both years (right): The blue horizontal line indicates the suggestive threshold -$\log_{10}$ (8.98E-7).

**Source data 5.** Quantile-quantile plots of GWAS in 2 years, 2018 (left) and 2019 (center), and best linear unbiased estimates (right).

**Figure supplement 1.** Overview of the field experiment and exemplary images demonstrating phenotypic traits.

**Figure supplement 2.** Graphical presentation of correlations between years among 12 traits.

**Figure supplement 3.** Pearson correlations among 12 quinoa traits.

**Figure supplement 4.** Principal component analysis (PCA) of 12 quantitative phenotypes.

**Figure supplement 5.** Principal component analysis (PCA) of four quantitative traits (days to flowering, days to maturity, plant height, and panicle length).

**Figure supplement 6.** GWAS analysis of principal components, PC1 (**a**), PC2 (**b**), PC3 (**c**): Manhattan plots (left), and quantile-quantile plots (right).

**Figure supplement 7.** The effect of repetitive sequences on identification of marker-trait associations in quinoa.

within this region and three of these were without known functions. The most significant SNP is located within the *CqGLX2-2* gene, which encodes an enzyme of the glyoxalase family (*Figure 3*). The allele carrying a cytosine at the position with the most significant SNP is associated with early flowering, maturing, and short panicles and plants (*Figure 3b*). These traits are essential for the adaptation to long-day conditions.

TSW is an important yield component. We found a strong MTA between 63.2 and 64.87 Mb on chromosome Cq8B. Significantly associated SNPs were localized within two genes (*Figure 4*). One gene displays homology to *PP2C*, encoding a member of the phosphatase-2C (*PP2C*) family protein and the second gene encodes a member of the RING-type E3 ubiquitin ligase family. These genes were found to be involved in controlling seed size in soybean, maize, rice, soybean, and *Arabidopsis* (*Li et al., 2019*). We then checked haplotype variation and identified five and seven haplotypes for *CqPP2C* and *CqRING* genes, respectively. Accessions carrying PP2C_hap3 and RING_hap7 displayed larger seeds in both years (*Figure 4* and *Figure 4—figure supplement 2*).

Downy mildew is one of the major diseases in quinoa, which causes massive yield damage. Notably, our GWAS identified strong MTA for resistance against this disease. The most significant SNPs are located in subgenome A (*Figure 3—source data 4*). Thus, the A-genome progenitor seems to be the donor of downy mildew resistance. In addition, we identified a candidate gene within a region 38.99–39.03 Mb on chromosome Cq2A, which showed the highest significant association (*Figure 3—source data 4*). This gene encodes a protein with an NBS-LRR (nucleotide-binding site leucine-rich repeat) domain often found in resistance gene analogs with a function against mildew infection (*Zhang et al., 2019a*).

## Discussion

We assembled a diversity set of 303 quinoa accessions and 7 accessions from wild relatives. Plants were grown under northern European conditions, and agronomically important traits were studied. In total, 2.9 million SNPs were found after re-sequencing. We found substantial phenotypic and genetic variation. Our diversity set was structured into two highly diverged populations, and genomic regions associated with this diversity were localized. Due to a high marker density, candidate genes controlling qualitative and quantitative traits were identified. The high genetic diversity and rapid LD breakdown reflect the short breeding history of this crop.

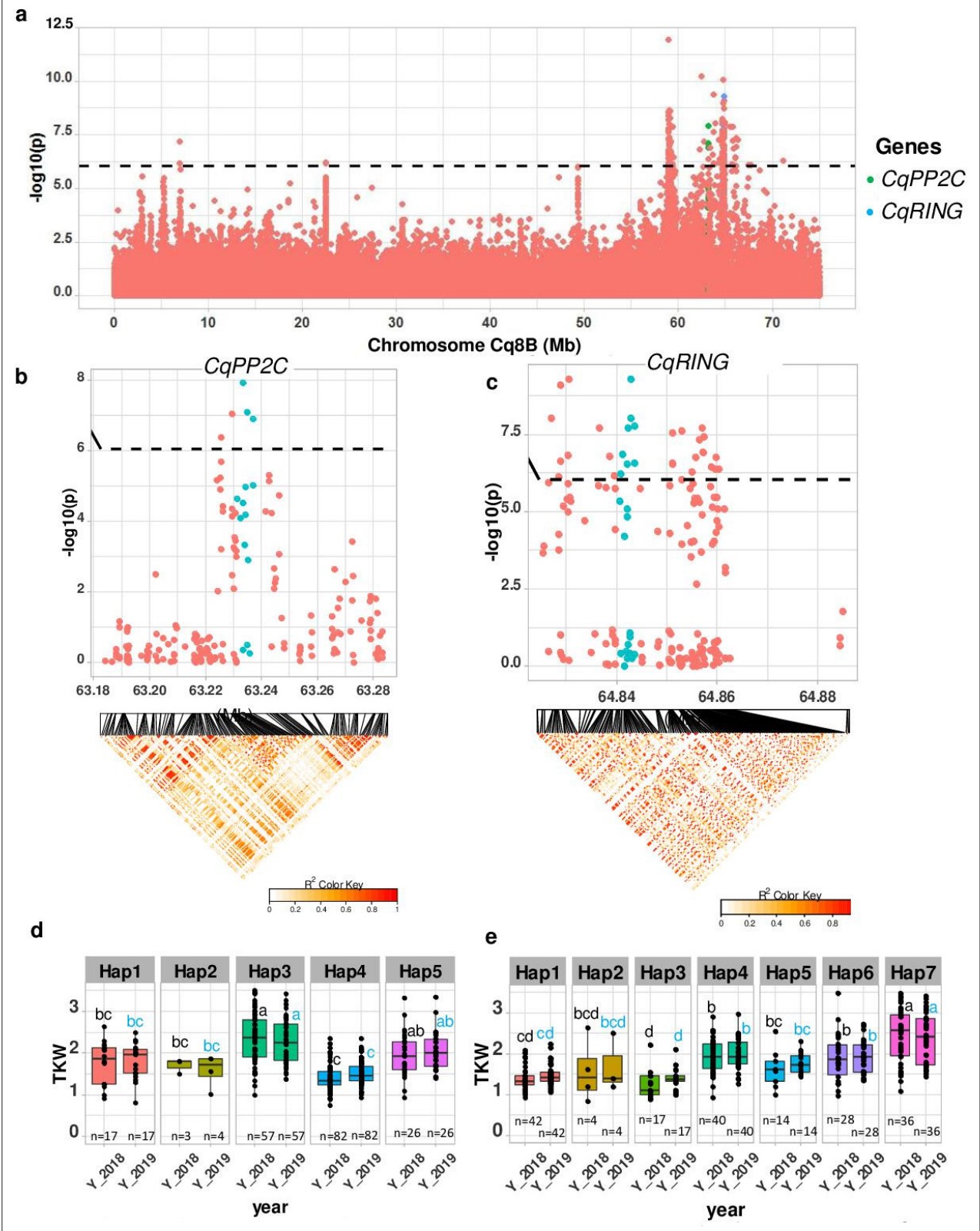

**Figure 4.** Identification of candidate genes for thousand seed weight. (**a**) Manhattan plot from chromosome Cq8B. Green and blue dots depict the *CqPP2C5* and the *CqRING* gene, respectively. (**b**) Top: Local Manhattan plot in the neighborhood of the *CqPP2C* gene. Bottom: Linkage disequilibrium (LD) heatmap. (**c**) Top: Local Manhattan plot in the neighborhood of the *CqRING* gene. Bottom: LD heatmap. Differences in thousand seed weight

*Figure 4 continued on next page*

*Figure 4 continued*

between five *CqPP2C* (**d**) and seven *CqRING* haplotypes (**e**). ANOVA was performed for each year separately to determine significance among haplotypes and grouping of pairwise multiple comparison was obtained using Tukey's test. (blue and black colour letters are for different years).

The online version of this article includes the following source data and figure supplement(s) for figure 4:

**Source data 1.** Manhattan plot from chromosome Cq8B.

**Source data 2.** Manhattan plot from chromosome Cq8B.

**Source data 3.** Differences in thousand seed weight between five *CqPP2C* (**d**) and seven *CqRING* haplotypes (**e**).

**Figure supplement 1.** Local Manhattan plots for (**a**) flower color, (**b**) saponin content, and (**c**) mildew infection.

**Figure supplement 2.** Haplotypes of two genes, *CqPP2C* (**a**) and *CqRING* (**b**), associated with seed size in quinoa.

We were aiming to assemble the first diversity set, which represents the genetic variation of this species. Therefore, we established a permanent resource that is genotypically and phenotypically characterized. We believe that this collection is important for future studies for the following reasons: we observed substantial phenotypic variation for all traits and high homogeneity within accessions. Moreover, low or absent phenotypic variation within accessions demonstrates homogeneity as expected for a self-pollinating species. Therefore, the sequence of one plant is representative of the whole accession, which is important for the power of the GWAS.

Today, over 16,000 accessions of quinoa are stored ex situ in seed banks in more than 30 countries (*Rojas et al., 2015*). Despite the enormous diversity, only a few accessions have been genotyped with molecular markers. We found a clear differentiation into Highland and Lowland quinoa. In previous studies, five ecotypes had been distinguished: Valley type, Altiplano type, Salar type, Sea level type, and Subtropical type (*Murphy, 2018*). Adaptation to different altitudes, tolerance to abiotic stresses such as drought and salt, and photoperiodic responses are the major factors determining ecotypes (*Gomez-Pando, 2015*). In our study, we could further allocate the quinoa accessions to five Highland and two Lowland subpopulations. This demonstrates the power of high-density SNP mapping to identity finer divisions at higher *K*. The origin of accessions and ecotype differentiation could be meaningfully interpreted by combining the information from phylogenetic data and population structure. As we expected, North American accessions (accessions obtained from USDA), clustered with Chilean accessions, suggesting sequence-based characterization of ecotypes would be more informative and reproducible. Moreover, high-density SNP genotyping unveiled the origin of unknown or falsely labeled gene bank accessions, as recently proposed by *Milner et al., 2019*. The geographical origin of 52 accessions from our panel was unknown. We suggest using phylogenic data and admixture results to complement the available passport data. For instance, two accessions with origin recorded as Chile are closely related to Peruvian and Bolivian accessions, which suggests that they also originate from Highland quinoa.

What can we learn about the domestication of quinoa and its breeding history by comparing our results with data from other crops? LD decay is one parameter reflecting the intensity of breeding. LD decay in quinoa (32.4 kb) is faster than in most studies with major crop species, for example, rapeseed (465.5 kb) (*Wu et al., 2019*), foxtail millet (*Setaria italica*, 100 kb) (*Jia et al., 2013*), pigeon pea (*Cajanus cajan*, 70 kb) (*Varshney et al., 2017a*), soybean (150 kb) (*Zhou et al., 2015*), and rice (200 kb) (*Mather et al., 2007*). Although comparisons must be regarded with care due to different numbers of markers and accessions, different types of reproduction, and the selection intensity, the rapid LD decay in quinoa reflects its short breeding history and low selection intensity. Moreover, quinoa is a self-pollinating species where larger linkage blocks could be expected. However, cross-pollination rates in some accessions can be up to 17.36% (*Silvestri and Gil, 2000*), which is exploited by small Andean farmers who grow mixed quinoa accessions to ensure harvest under different biotic and abiotic stresses. This may facilitate a certain degree of cross-pollination and admixture providing opportunities for cultivar improvement.

Interestingly, the LD structure between Highland and Lowland populations is highly contrasting (6.5 vs. 49.8 kb), indicating larger LD blocks in the Lowland population. Low nucleotide diversity and negative Tajima's *D* were also observed in the Lowland population compared to Highland quinoa. The population differentiation index and LD differences have been used to test the hypothesis of multiple domestication events. As an example, different domestication bottlenecks have been reported for *japonica* (LD decay: 65 kb) and *indica* rice (LD decay: 200 kb) (*Xu et al., 2011*). The estimated $F_{ST}$

value from this study (0.36) is in the similar range of $F_{ST}$ estimates in rice subspecies *indica* and *japonica* (0.55) (*Huang et al., 2010*) and melon (*Cucumis melo*) subspecies *melo* and *agrestis* (0.46) (*Zhao et al., 2019*). Two hypotheses have been proposed for the domestication of quinoa from *C. hircinum*: (1) one event that gave rise to Highland quinoa and subsequently to Lowland quinoa and (2) two separate domestication events giving rise to Highland and Lowland quinoa independently (*Jarvis et al., 2017*). However, our study does not follow the second hypothesis because *C. hircinum* accession BYU 566 was basal to both clades of the phylogenetic tree (Highland and Lowland). Moreover, our wild *Chenopodium* germplasm does not represent enough diversity for in-depth analysis of domestication events. Therefore, we propose three possible scenarios to explain significant differences in LD structure, nucleotide diversity, Tajima's $D$ and $F_{ST}$: (1) two independent domestication events with a substantial bottleneck on Lowland populations, (2) single domestication but strong population growth after adaptation of Lowland quinoa, or (3) strong adaptive selection after domestication. To understand the history and genetics of domestication, it will be necessary to sequence a large representative set of outgroup species such as *C. berlandieri*, *C. hircinum, C. pallidicaule,* and *C. suecicum*.

Apart from marker density and sample size, the power of GWAS depends on the quality of the phenotypic data. Plants were grown in Northern Europe. Therefore, the MTAs are, first of all, relevant for temperate long-day climates. The share of genetic variances and, thus, the heritabilities were high across environments. We expect higher genotype × environment interaction for flowering time, DTM, PH, and PL if short-day environments will be included because many accessions have a strong day-length response (data not shown). Differentiation of germplasm from different environments may be very strong and therefore bias the GWAS results. Nonetheless, all accessions reached the flowering stage and produced seeds. Therefore, bias toward maturity is a consequence of environmental adaptation or lack of adaptation to long-day conditions.

Furthermore, the genes controlling Mendelian traits precisely coincided or were in LD with significant SNP positions, as exemplified by the genes associated with saponin content and flower color. Hence, the diversity panel provides sufficient power to identify SNP-trait associations for important agronomic traits such as TSW and downy mildew tolerance. In different plant species, seed size is controlled by six different pathways (*Li et al., 2019*). We found two important genes associated with seed size from the Brassinosteroid (*CqPP2C*) and the ubiquitin-proteasome (*CqRING*) pathway. The non-functional allele of soybean *PP2C1* is associated with small seeds (*Lu et al., 2017*). We detected a superior haplotype (PP2C_hap3), which is associated with larger seeds. *CqRING* encodes an E3 ubiquitin ligase protein. There are two RING-type E3 ubiquitins known as *DA1* and *DA2,* involved in the seed size controlling pathway. They were found in *Arabidopsis* rice, maize, and wheat. Downy mildew is the most acute disease for quinoa, caused by the fungus *Peronospora variabili* (*Choi et al., 2010*). A recent study attempted the identification of genes based on a GWAS analysis in quinoa. However, no significant associations were found, probably due to the lack of power because of the low number of accessions used (61 and 88) (*Colque-Little et al., 2021*); nonetheless, the most significant SNPs of that study were comparable to the findings of the present study. In our study, a strong MTA provides preliminary indications that the NBS-LRR gene on chromosome Cq2A contributes to downy mildew resistance in quinoa. We propose using this sequence for marker-assisted selection in segregating $F_2$ populations produced during pedigree breeding, after the confirmation of its role in downy mildew resistance by functional analysis in quinoa.

In this study, the advantage of multivariate analysis of cross-phenotype association became obvious. We identified candidate genes with pleiotropic effects on days to flowering, DTM, PH, and PL. Genes involved in the flowering time pathway have been shown to have pleiotropic effects (*Auge et al., 2019*). However, flowering time is a major developmental stage that can influence other agronomical traits like PH and PL. Therefore, the correlation among traits may have led to the association of flowering time genes with multiple traits and should be further investigated. Interestingly, the most significant and consistent SNP association was residing within a putative *GLX-2* ortholog. *GLX* genes, among other functions, have impacted cell division and proliferation in *Amaranthus paniculatus* (*Chakravarty and Sopory, 1998*). Therefore, the *CqGLX-2* gene is one putative candidate for controlling day-length response. Furthermore, the genes identified within these regions that encode proteins with unknown functions should be subjected to future studies.

This study also has a major breeding perspective. We aimed to elucidate the potential of quinoa for cultivation in temperate climates. Evidently, many accessions are not adapted to northern European

climate and photoperiod conditions because they flowered too late and did not reach maturity before October. Nevertheless, 48 accessions are attractive as crossing partners for breeding programs because they are insensitive to photoperiod or long-day responsive. Moreover, they are attractive due to their short PH, low tillering capacity, favorable inflorescence architecture, and high TSW. These are important characters for mechanical crop cultivation and combine harvesting. The MTA found in this study offers a perspective to use parents with superior phenotypes in crossing programs. We suggest a genotype-building strategy by pyramiding favorable alleles (haplotypes). In this way, also accessions from our diversity set, which are not adapted to long-day conditions but with favorable agronomic characters, will be considered. Then, favorable genotypes will be identified from offspring generations by marker-assisted selection using markers in LD with significant SNPs. Furthermore, the MTA from this study will be valid for allele mining in quinoa germplasm collections to identify yet unexploited genetic variation.

## Materials and methods

### Plant materials and growth conditions

We selected 350 quinoa accessions for phenotyping, and of these, 296 were re-sequenced in this study. Re-sequencing data of 14 additional accessions that had already been published (*Jarvis et al., 2017*) were also included in the study, together with the wild relatives (*C. belandieri* and *C. hircinum*) (*Jarvis et al., 2017*). These accessions represent different geographical regions of quinoa cultivation (*Supplementary file 1a*). Plants were grown in the field in Kiel, Northern Germany, in 2018 and 2019. (The local weather data is provided in *Supplementary file 1h*.) Seeds were sown in the second week of April in 35× multi-tray pots. Then, plants were transplanted to the field in the first week of May as single-row plots in a randomized complete block design with three blocks. The distances between rows and between plants were set to 60 and 20 cm, respectively. Each row plot contained seven plants per accession.

We recorded days to bolting (days to bolting) as BBCH51 and days to flowering as BBCH60 twice a week during the growth period. DTM was determined when plants reached complete senescence (BBCH94)(*Stanschewski et al., 2021*). If plants did not reach this stage, DTM was set as 250 days. In both years, plants were harvested in the second week of October. PH, PL, and the NoB were phenotyped at harvest. STL (*Figure 3—figure supplement 1*) was scored on a scale from 1 to 5, where score 1 indicates no STL. Similarly, panicle density was recorded on a scale from 1 to 7, where density 1 represents lax panicles, and panicle density 7 represents highly dense panicles. Flower color and stem color were determined by visual observation. Pigmented and non-pigmented plants were scored as 1 and 0, respectively. GT was classified into two categories and analyzed as a dichotomous trait as well. We observed severe mildew infection in 2019. Therefore, we scored mildew infection on a scale from 1 to 3, where 1 equals no infection, and 3 equals severe infection.

### Statistical analysis

We calculated the BLUE of the traits across years by fitting a linear mixed model using the lme4 R package (*Bates et al., 2015*). We used the following model:

### Genome sequencing and identification of genomic variations

For DNA extraction, two plants per genotype were grown in a greenhouse at the University of Hohenheim, and two leaves from a single 2-month-old plant were collected and frozen immediately. DNA was subsequently extracted using the AX Gravity DNA extraction kit (A&A Biotechnology, Gdynia, Poland) following the manufacturer's instructions. The purity and quality of DNA were controlled by agarose gel electrophoresis and the concentration was determined with a Qubit instrument using SYBR green staining. Whole-genome sequencing was performed for 312 accessions at Novogene (China) using short-reads Illumina NovaSeq S4 Flowcell technology and yielded an average of 10 Gb of paired-end (PE) 2 × 150 bp reads with quality Q>30 Phred score per sample, which is equivalent to ~7× coverage of the haploid quinoa genome (~1.45 Gb). We then used an automated pipeline compiled based on the Genome Analysis Toolkit (*Abrouk et al., 2020*). Raw sequence reads were filtered with trimmomatic-v0.38 (*Bolger et al., 2014*) using the following criteria: LEADING:20; TRAILING:20; SLIDINGWINDOW:5:20; MINLEN:50. The filtered paired-end reads were then individually mapped for each

sample against an improved version of the QQ74 quinoa reference genome (CoGe id60716) using BWA-MEM (v-0.7.17) (*Li and Durbin, 2010*) followed by sorting and indexing using samtools (v1.8) (*Li et al., 2009*). Duplicated reads were marked, and read groups were assigned using the Picard tools (http://broadinstitute.github.io/picard/). Variants were identified with GATK (v4.0.1.1) (*McKenna et al., 2010*; *Van der Auwera et al., 2013*) using the '--emitRefConfidence' function of the Haplo-typeCaller algorithm and '—heterozygosity' value set at 0.005 to call SNPs and InDels for each accession. Individual g.vcf files for each sample were then compressed and indexed with tabix (v-0.2.6) (*Li, 2011*) and combined into chromosome g.vcf using GenomicsDBImport function of GATK. Joint genotyping was then performed for each chromosome using the function GenotypeGVCFs of GATK. To obtain high confidence variants, we excluded SNPs with the VariantFiltration function of GATK with the criteria: QD < 2.0; FS > 60.0; MQ < 40.0; MQRankSum < −12.5; ReadPosRankSum < −8.0 and SOR > 3.0. Then, SNP loci which contained more than 70% missing data, were filtered by VCFtools (*Danecek et al., 2011*) (v0.1.5), which resulted in our initial set of ~45 M SNPs for all the 332 accessions, including 20 previously re-sequenced accessions (*Jarvis et al., 2017*). All re-sequencing data are submitted to SRA under project id BioProject PRJNA673789.

$$Y_{ikj} = \mu + \text{Accession}_i + \text{Block}_i + \text{Year}_i + (\text{Accession} \times \text{Block})_{ij} + (\text{Accession} \times \text{Year})_{ik} + \text{Error}_{ijk}$$

where $\mu$ is the mean, Accession$_i$ is the genotype effect of the $i$th accession, Block$_j$ is the effect of the $j$th Block, Year$_k$ is the effect of the $k$th year, (Accession × Block)$_{ij}$ is the Accession-Block interaction effect, Accession × Year$_{ik}$ is the accession-year interaction effect, Error$_{ijk}$ is the error of the $j$th block in the $k$th year. We treated all items as random effects for heritability estimation, and for BLUE, accessions were treated as fixed effects. We analyzed the PCs of phenotypes using the R package FactoMineR (*Lê et al., 2008*).

In our panel, we had three triplicates for quality checking and nine duplicates between *Jarvis et al., 2017*, and 312 newly re-sequenced accessions. In order to remove duplicates, as a preliminary analysis, we removed SNP loci with a minimum mean depth < 5 across samples and SNP loci with more than 5% missing data. Then, we filtered SNPs with a minor allele frequency lower than 0.05 (MAF < 0.05). After these filtering steps, we obtained a VCF file that contained 229,017 SNPs. Then, we construct a ML tree. First, we used the modelFinder (*Kalyaanamoorthy et al., 2017*) in IQ-TREE v1.6.619 (*Nguyen et al., 2015*) to determine the best model for ML tree construction. We selected GTR + F + R8 (GTR: General time-reversible, F: Empirical base frequencies, R8: FreeRate model) as the best fitting model according to the Bayesian information criterion (BIC) estimated by the software. Next, we used 1000 replicates with ultrafast bootstrapping (UFboots) (*Hoang et al., 2018*) to check the reliability of the phylogenetic tree. To visualize the phylogenetic tree, we used the Interactive Tree Of Life tool (https://itol.embl.de/) (*Letunic and Bork, 2016*). Then, based on the phylogenetic tree, we removed duplicate accessions and accessions with unclear identities. After the quality control, we retained 310 accessions (303 quinoa accessions and 7 wild *Chenopodium* accessions).

Then, we used the initial SNP set and defined two subsets using the following criteria: (1) A base SNP set of 5,817,159 biallelic SNPs obtained by removing SNPs with more than 50% missing geno-type data, minimum mean depth less than five, and minor allele frequency less than 1%. (2) A high confidence (HCSNP) set of 2,872,935 SNPs from the base SNP set was created by removing SNPs with a minor allele frequency of less than 5%. We used custom scripts for dividing homozygous and hetero-zygous SNP regions of each sample. Then, using custom bash scripts, mean DP and GQ calculations were carried out. The base SNP set was used for the diversity statistics, and the high confidence SNPs set was used for GWAS analysis. To obtain the number of population-specific SNPs, we applied 5% MAF separately for each population.

We annotated the high confidence SNP using SnpEff 4.3T (*Cingolani et al., 2012a*) and a custom database (*Cingolani et al., 2012b*) based on the QQ74 reference genome and annotation (CoGe id60716). Afterward, we extracted the SNP annotations using SnpSift (*Cingolani et al., 2012a*). Based on the annotations, SNPs were mainly categorized into five groups: (1) upstream of the transcript start site (5 kb), (2) downstream of the transcript stop site (5 kb), (3) coding sequence (CDS), (4) intergenic, and (5) intronic. We used SnpEff to categorize SNPs in coding regions based on their effects: synon-ymous, missense, splice acceptor, splice donor, splice region, start lost, start gained, stop lost, and spot retained.

### Phylogenetic analysis and population structure analysis

For population structure analysis, we employed SNP subsets, as demonstrated in previous studies, to reduce the computational time (*Wang et al., 2018*). We created 10 randomized SNP sets, each containing 50,000 SNPs. First, the base SNP set was split into 5000 subsets of an equal number of SNPs to create subsets. Then, 10 SNPs from each subset were randomly selected, providing a total of 50,000 SNPs in a randomized set (randomized 50k set). We then repeated this procedure for nine more times and finally obtained 10 randomized 50k sets. Population structure analysis was conducted using ADMIXTURE (version 1.3) (*Alexander et al., 2009*). We ran ADMIXTURE for each subset separately with a predefined number of genetic clusters *K* from 2 to 10 and varying random seeds with 1000 bootstraps. Also, we performed the cross-validation procedure for each run. Obtained *Q* matrices were aligned using the greedy algorithm in the CLUMPP software (*Jakobsson and Rosenberg, 2007*). Population structure plots were created using custom R scripts. We then combined SNP from the 10 subsets to create a single SNP set of 434,077 unique SNPs for the phylogenetic analysis. We used the same method mentioned above to create the phylogenetic tree. Here, we selected the model GTR + F + R6 based on the BIC estimates. For the PCA, we used the high confidence SNP set and analysis was done in R package SNPrelate (*Zheng et al., 2012*). We estimated the top 10 PCs. The first (PC1) and second (PC2) were plotted using custom R scripts.

### Genomic patterns of variation

Using the base SNP set, we calculated nucleotide diversity ($\pi$) for subpopulations and $\pi$ ratios for Highland and Lowland population regions with the top 1% ratios of $\pi_{Highland}$/ $\pi_{Lowland}$ candidate regions for population divergence. We also estimated Tajima's *D* values for both populations to check the influence of selection on populations. $F_{ST}$ values were calculated between Highland and Lowland populations using the 10 kb non-overlapping window approach. Nucleotide diversity, Tajima's *D*, and $F_{ST}$ calculations were carried out in VCFtools (v0.1.5) (*Danecek et al., 2011*). We also performed a local PCA using the template of the lostruct program (*Li and Ralph, 2019*) to identify genomic regions with strong population structures. The genome was divided into 50 kb non-overlapping windows, and a PCA was conducted for each window using lostruct. We obtained candidate regions based on the local PCA (top 1% threshold). Genes located in these regions were considered as candidate genes underlying Highland and Lowland population divergence.

### LD analysis

First, we calculated LD in each population separately (Highland and Lowland). Then, LD was calculated in the whole population, excluding wild accessions. For LD calculations, we further filtered the high confidence SNP set by removing SNPs with >80% missing data (*Varshney et al., 2019*). Using a set of 2,513,717 SNPs, we calculated the correlation coefficient ($r^2$) between SNPs up to 300 kb apart by setting -MaxDist 300 and default parameters in the PopLDdecay software (*Zhang et al., 2019b*). Pairwise distance at $r^2$=0.20 is defined as the threshold for genome-wide LD decay. LD decay was plotted using custom R scripts based on the ggplot2 package.

### Genome-wide association study

We used the BLUE of traits and high confidence SNPs for the GWAS analysis. Morphological traits were treated as dichotomous traits and analyzed using generalized mixed linear models with the lme4 R software package (*Bates et al., 2015*). We used population structure and genetic relationships among accessions to minimize false-positive associations. Population structure represented by the PC was estimated with the SNPrelate software (*Zheng et al., 2012*). Genetic relationships between accessions were represented by a kinship matrix calculated with the efficient mixed-model association expedited (EMMAX) software (*Kang et al., 2010*) using high confidence SNPs (*Figure 2—figure supplement 1*). Then, we performed an association analysis using the mixed linear model for the whole population, including *K* and *P* matrices in EMMAX. We estimated the effective number of SNPs (*n*=1,062,716) using the Genetic type I Error Calculator (GEC) (*Li et al., 2012*) and calculated quantile-quantile plots (*Figure 3—source data 5*). We set the significant p-value threshold (Bonferroni correction, 0.05 /*n*, -log$_{10}$(4.7e-08)=7.32) and suggestive significant threshold (1 /*n*, -log$_{10}$(9.41e-7)=6.02) to identify significant loci underlying traits. Next, we checked how repetitive sequences influence the GWAS analysis. Here, we used repeat masker to exclude all the SNPs that are located in repetitive

regions of the genome. After removing SNPs on repetitive sequences, we obtained 1,906,734 SNPs and GWAS was carried out following the same method explained previously. We plotted SNP p-values on Manhattan plots using the qqman R package (*Turner, 2014*).

## Acknowledgements

We thank David Jarvis for providing the updated version of the quinoa reference genome. We thank Monika Bruisch, Brigitte Neidhardt-Olf, Elisabeth Kokai-Kota, Verena Kowalewski, and Gabriele Fiene for technical assistance. The financial support of this work was provided by the Competitive Research Grant (Grant No. OSR-2016-CRG5-466 2966-02) of the King Abdullah University of Science and Technology, Saudi Arabia, and baseline funding from KAUST to Mark Tester. We acknowledge financial support by DFG within the funding programme "Open Access Publikationskosten" to cover the article processing charges.

## Additional information

### Funding

| Funder | Grant reference number | Author |
|---|---|---|
| King Abdullah University of Science and Technology | OSR-2016-CRG5- 466 2966-02 | Dilan SR Patiranage |

The funders had no role in study design, data collection and interpretation, or the decision to submit the work for publication.

### Author contributions

Dilan SR Patiranage, Data curation, Formal analysis, Investigation, Methodology, Visualization, Writing – original draft; Elodie Rey, Resources, Data curation, Formal analysis, Validation, Visualization, Writing – review and editing; Nazgol Emrani, Conceptualization, Supervision, Investigation, Visualization, Writing – review and editing; Gordon Wellman, Sandra M Schmöckel, Resources, Writing – review and editing; Karl Schmid, Resources, Visualization, Writing – review and editing; Mark Tester, Conceptualization, Resources, Supervision, Funding acquisition, Visualization, Project administration, Writing – review and editing; Christian Jung, Conceptualization, Supervision, Funding acquisition, Visualization, Writing – review and editing

### Author ORCIDs

Dilan SR Patiranage http://orcid.org/0000-0001-6308-1838
Nazgol Emrani http://orcid.org/0000-0001-5673-3957
Karl Schmid http://orcid.org/0000-0001-5129-895X
Mark Tester http://orcid.org/0000-0002-5085-8801
Christian Jung http://orcid.org/0000-0001-8149-7976

### Decision letter and Author response

Decision letter https://doi.org/10.7554/eLife.66873.sa1
Author response https://doi.org/10.7554/eLife.66873.sa2

## Additional files

### Supplementary files

• Supplementary file 1. Quinoa accessions, growth conditions, high confidence SNP identification and marker-trait associations and candidate genes identified in the current study. (a) Accessions from the quinoa diversity panel and results from re-sequencing. (b) Summary of high-quality SNPs identified in quinoa accessions. (c) Variance components analysis of 12 quantitative traits. (d) Summary of marker-trait associations (MTA). (e) Candidate genes linked to SNPs with significant trait associations. (f) Summary of MTA associated with days to flowering, days to maturity, panicle density, and plant height identified on chromosome Cq2A. (g) Candidate genes located within the 50 kb flanking regions of significantly associated SNPs from the multivariate GWAS analysis. (h) Daily

temperature, precipitation, humidity, radiation, and wind speed during the cultivation seasons 2018 and 2019 in Kiel, Germany.

- Transparent reporting form

### Data availability

The raw sequencing data have been submitted to the NCBI Sequence Read Archive (SRA) under the BioProject PRJNA673789. Quinoa reference genome version 2 is available at CoGe database under genome id 53523. Phenotype data and ready-use genotype data (vcf file) are available at https://doi.org/10.5061/dryad.zgmsbcc9m. A detailed description of the germplasm, phenotyping methods, and phenotypes are available at https://quinoa.kaust.edu.sa/#/ (Stanschewski et al., 2021). Seeds are available from the public gene banks such as IPK Gatersleben (https://www.ipk-gatersleben.de/en/genebank/) and the U.S. National Plant Germplasm System (https://npgsweb.ars-grin.gov/grin-global/search). Custom scripts used for SNP calling are available on GitHub: https://github.com/IBEXCluster/IBEX-SNPcaller/blob/master/workflow.sh (copy archived at swh:1:rev:dad4095515fdb-80f390bee8893bb6f4453e12a73). Additional information of other custom scripts used for making plots are available at https://github.com/DilanSarange/quinoaDPgwas, (copy archived at swh:1:rev:4e68ba3c64f3e67d156883c6c40dbd4d3701839c).

The following datasets were generated:

| Author(s) | Year | Dataset title | Dataset URL | Database and Identifier |
|---|---|---|---|---|
| Patiranage DSR, Rey E, Emrani N, Wellman G, Schmid K, Schmöckel SM, Tester M, Jung C | 2021 | Genome-wide association study in quinoa reveals selection pattern typical for crops with a short breeding history | https://doi.org/10.5061/dryad.zgmsbcc9m | Dryad Digital Repository, 10.5061/dryad.zgmsbcc9m |
| Jung C | 2022 | Chenopodium quinoa diversity genome re-sequencing | https://www.ncbi.nlm.nih.gov/bioproject/PRJNA673789 | NCBI BioProject, PRJNA673789 |

The following previously published dataset was used:

| Author(s) | Year | Dataset title | Dataset URL | Database and Identifier |
|---|---|---|---|---|
| Jarvis DE, Ho YS, Lightfoot DJ, Schmöckel SM, Li B, Borm TJA, Ohyanagi H, Mineta K, Michell CT, Saber N, Kharbatia NM, Rupper RR, Sharp AR, Dally N, Boughton BA, Woo YH, Gao G, Schijlen E, Guo X, Momin AA, Negrão S, Al-Babili S, Gehring C, Roessner Ute, Jung C, Murphy K, Arold ST, Gojobori T, Gerard van der Linden C, van Loo EN, Jellen EN, Maughan PJ, Tester M | 2017 | The genome of Chenopodium quinoa | http://www.cbrc.kaust.edu.sa/chenopodiumdb/ | Phytozome database, chenopodiumdb |

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
