## [Editor Report]

This is a comprehensive study of genomic and phenotypic diversity in the orphan crop quinoa. Based on whole genome resequencing of 310 accessions and field phenotyping of the same set of accessions for two years, the study identified the genetic basis of agronomically important traits. Based on this promising work, there will likely be scope for quick improvement of this orphan crop through breeding.

---

## [Decision Letter]

**Decision letter after peer review:**

Thank you for submitting your article "Genome-wide association study in quinoa reveals selection pattern typical for crops with a short breeding history" for consideration by *eLife*. Your article has been reviewed by 3 peer reviewers, and the evaluation has been overseen by Vincent Castric as the Reviewing Editor and Meredith Schuman as the Senior Editor. The following individuals involved in review of your submission have agreed to reveal their identity: Justin O Borevitz (Reviewer #2); Stig Uggerhøj Andersen (Reviewer #3).

Essential revisions:

The reviewers appreciated the useful resource for quinoa and competent analysis of the diversity in this orphan crop. Please find below a list of essential revisions that need to be addressed before we can consider inclusion of this work in *eLife*. The public peer reviews are also appended for your information and for eventual inclusion alongside your preprint.

(1) There was a consensus that the population genomic analyses do not meet the potential of the dataset and should be further developed; while a good descriptive assessment of quinoa, the results do not allow a larger impact. For instance, an Fst scan could pinpoint candidates for local adaptation between lowland and highland quinoa; local PCA is another option suggested by Reviewer 2, https://pubmed.ncbi.nlm.nih.gov/30459280/. A possibility to substantiate the claim that specific traits are relevant for the differentiation of highland and lowland varieties, could be to combine these Fst scans with GWAS and look for skews in Fst distributions.

(2) To be of more general interest, it would be good to see what the GWAS tells us about evolution of this crop species. Can the authors see if it is associated with selection, maybe in highland vs. lowland? Separate GWAS between the highland vs lowland varieties could be used to check whether the same alleles are controlling traits across subpopulations. Besides the temporal replication, this could provide important information on the replicability of the associations across groups. It is possible that different associations are supported in these two different groups. Specific comments from Reviewer 2:

– For GWAS across the major subpopulations (highland/lowland), a joint analysis is only helpful when the same alleles are controlling traits across subpopulations. Is this the case? How much variation does the kinship matrix explain for the traits? What does the histogram (pairwise accession distances) of kinship look like? If 2 clear groups perhaps separate analysis is preferred.

– In particular DTM, days to maturity, could be highly confounded with highland/lowland ecotype. The multi year field trial data is a very strong part of this paper with direct agronomic relevance, but were the growing conditions typical of lowland, longer season conditions, that prevented many accession from reaching maturity? Many yield traits depend on maturity time and could vary jointly with DTM, eg multitrait analysis or a regression of yield on DTM. You investigate this with PCA(CP), but I don't find this unsupervised approach informative, and it could be excluded.

(3) For a resources paper, it is especially important to clarify the way the data and accessions are made available to the community. To ensure accessibility of the accessions, seeds should ideally be deposited in a genebank, where they are propagated and can be ordered online. Otherwise, it might be difficult for others to get hold of the seeds in practice. If that is not possible, please specifically state that the seeds can be obtained from the authors. The genomic data should ideally be made available through a public genome browser, or alternatively deposited in appropriate databases (including the vcf files).

(4) There was also a consensus that the description of the candidate genes was too assertive, as the associations do not demonstrate causality at this stage. This should be toned down.

5) Regarding SNP calling, reviewer 3 made important suggestions to evaluate robustness of the results to a number of potential pitfalls (coverage and repetitive regions):

– The correlation between coverage and heterozygosity levels represents a serious issue that should be addressed. It seems that you need more than 6x coverage to achieve accurate calling. The obvious solution would be to carry out additional sequencing, but this would cause significant delays. Another option would be to scrutinize the information in the VCF files. Many SNP-callers, including GATK, tend to err on the side of caution and call heterozygotes rather than homozygotes in difficult cases. You may well be able to salvage accurate data from the low-coverage individuals by adjusting genotype calls based on the genotype likelihoods. You could require compelling evidence for a heterozygote before it is called, instead favoring the most like homozygous genotype. Please provide a scatter plot of depth versus heterozygosity level to demonstrate that the issue was resolved.

– Although you have very reasonably applied a mapping quality filter to reduce the problem of poor-quality SNPs derived from repetitive regions, problems could persist. If I read Table S5 correctly, there is an overwhelming majority of intergenic SNPs. It would be interesting to see what happens if you eliminate SNPs in repetitive regions, as determined by RepeatMasker or similar, and then rerun the GWAS. If the repetitive regions contribute false positive SNPs with random genotype calls, you should get cleaner results with more significant associations.

– A specific example is lines 218-227, where you argue for mapping resolution. The argument does not appear strong as none of the top SNPs in the region are located in the two candidate genes according to Figure S14b. Also it is unclear from that figure how many other genes reside within the region. It would be preferable to use the extent of LD compared to gene density as a general argument for mapping resolution.

[Editors’ note: further revisions were suggested prior to acceptance, as described below.]

Thank you for resubmitting your work entitled "Genome-wide association study in quinoa reveals selection pattern typical for crops with a short breeding history" for further consideration by *eLife*. Your revised article has been evaluated by Vincent Castric as Reviewing editor and the evaluation has been overseen by Meredith Schuman as Senior Editor.

The manuscript has been improved but there are some remaining issues that need to be addressed, as outlined below:

1 – The section on « genomic patterns of variations between highland and lowland quinoa » has been expanded, but unfortunately at this stage it remains too descriptive. We understand from your response letter that a more comprehensive analysis of the demographic and selective history of quinoa is in preparation, so we suggest that this section could be shortened to include only : (1) the strong population differentiation between highland and lowland accessions, (2) the comparison of LD decay, nucleotide diversity and Tajima's D between highland and lowland, (3) the local PCA and a comparison of average FST across chromosomes to test for heterogeneity of patterns of differentiation. e.g. Do some chromosomes show a lower FST and different contribution to the three corners of the PCA, in particular Cq6B, as would be expected if it experienced recent introgression?

2 – The issue raised by reviewer 3 that the modest sequencing depth could lead to inaccurate SNP calling was correctly addressed by plotting mean heterozygosity vs sequencing depth across the 310+ accessions, but it is incomplete. The reviewer suggested that the genotype likelihood threshold to call heterozygous sites could be adjusted for each accession by examining the mean sequencing depth and mean genotype likelihood between heterozygous and homozygous sites. The eventual difference of means between accessions with high vs low overall coverage could be used to adjust the threshold.

3 – Along the same lines, a missing piece of basic information is the proportion of SNPs called as homozygous vs heterozygous. Given the high selfing rate, a high proportion of homozygous SNPs is expected. Is this observed, and does that vary across accessions?

4 – Finally, reviewer 3 had also suggested restricting the set of SNPs to those outside repeated regions, as identified by e.g. RepeatMasker in the reference genome. Even using stringent filters to identify SNPs will not totally alleviate the problem of SNP calling in repeated regions, and these will remain dubious. Please do consider testing whether the marker-trait associations detected are still detected (and perhaps more convincingly so) when removing SNPs in repeated regions.

5 – The text contains too many acronyms that are rarely useful. Examples of acronyms that are not repeated frequently and could make the text a lot easier to follow if the words were instead written in full include : HCSNPs, NoB, STL, DTM, PL, PH, TSW, GT, DTB, DTF, PCA(PHEN), BLUE, MTA, BBCH60, BBCH94.

---

## [Author Response]

Essential revisions:1) There was a consensus that the population genomic analyses do not meet the potential of the dataset and should be further developed; while a good descriptive assessment of quinoa, the results do not allow a larger impact. For instance, an Fst scan could pinpoint candidates for local adaptation between lowland and highland quinoa; local PCA is another option suggested by Reviewer 2, https://pubmed.ncbi.nlm.nih.gov/30459280/. A possibility to substantiate the claim that specific traits are relevant for the differentiation of highland and lowland varieties, could be to combine these Fst scans with GWAS and look for skews in Fst distributions.

Thanks for this suggestion. We performed the suggested analysis by dividing the genome in 50 kb windows, and the results of the analysis are presented in Figure 1—figure supplement 8. The results are added to the text, lines 184-205 and 544-549.

2) To be of more general interest, it would be good to see what the GWAS tells us about evolution of this crop species. Can the authors see if it is associated with selection, maybe in highland vs. lowland? Separate GWAS between the highland vs lowland varieties could be used to check whether the same alleles are controlling traits across subpopulations. Besides the temporal replication, this could provide important information on the replicability of the associations across groups. It is possible that different associations are supported in these two different groups. Specific comments from Reviewer 2:

We checked the histogram of the kinship matrix as suggested in the following comments. This analysis did not support the hypothesis that there are distinct populations although it showed the high diversity within the quinoa panel. Therefore, we believe that the control for population stratification using PCA was adequate. Also, sub population-based studies tend to be underpowered, mainly when the number of genotypes is not big enough. Therefore, we did not implement the population-wise GWAS analysis. Moreover, we could find many studies of different plant species with highly diverged populations, where single GWAS analyses instead of separate analyses have been conducted, e.g.:

– Kang, L., Qian, L., Zheng, M., Chen, L., Chen, H., Yang, L., … and Liu, Z. (2021). Genomic insights into the origin, domestication and diversification of *Brassica juncea*. *Nature genetics*, *53*(9), 1392-1402.

– Varshney, R. K., Roorkiwal, M., Sun, S., Bajaj, P., Chitikineni, A., Thudi, M., … and Liu, X. (2021). A chickpea genetic variation map based on the sequencing of 3,366 genomes. *Nature*, 1-6.

– Wang, W., Mauleon, R., Hu, Z., Chebotarov, D., Tai, S., Wu, Z., … and Leung, H. (2018). Genomic variation in 3,010 diverse accessions of Asian cultivated rice. *Nature*, *557*(7703), 43-49.

– For GWAS across the major subpopulations (highland/lowland), a joint analysis is only helpful when the same alleles are controlling traits across subpopulations. Is this the case? How much variation does the kinship matrix explain for the traits? What does the histogram (pairwise accession distances) of kinship look like? If 2 clear groups perhaps separate analysis is preferred.

As mentioned in the previous reply, the histogram of the kinship coefficient (IBS) between all pairs of individuals does not show two groups; hence no separate analysis was performed ( Figure 2—figure supplement 1). A sentence is added (lines 230-232).

– In particular DTM, days to maturity, could be highly confounded with highland/lowland ecotype. The multi year field trial data is a very strong part of this paper with direct agronomic relevance, but were the growing conditions typical of lowland, longer season conditions, that prevented many accession from reaching maturity? Many yield traits depend on maturity time and could vary jointly with DTM, eg multitrait analysis or a regression of yield on DTM. You investigate this with PCA(CP), but I don't find this unsupervised approach informative, and it could be excluded.

We agree that the growing conditions typical of lowland (longer seasons) can prevent many accessions from reaching maturity. However, we observed that all accessions flowered and produced seeds. Nonetheless, GWAS with PCA (CP) has been shown to be effective in multiple studies (mentioned below) for genetically correlated traits. Therefore, we believe our analysis could address the bias that might occur due to maturity differences. We also discuss this in line 373-377.

– Miao, C., Xu, Y., Liu, S., Schnable, P. S., and Schnable, J. C. (2020). Increased power and accuracy of causal locus identification in time series genome-wide association in sorghum. Plant physiology, 183(4), 1898-1909.

– Yano, K., Morinaka, Y., Wang, F., Huang, P., Takehara, S., Hirai, T., … and Matsuoka, M. (2019). GWAS with principal component analysis identifies a gene comprehensively controlling rice architecture. *Proceedings of the National Academy of Sciences*, *116*(42), 21262-21267.

– Aschard, H., Vilhjálmsson, B. J., Greliche, N., Morange, P. E., Trégouët, D. A., and Kraft, P. (2014). Maximizing the power of principal-component analysis of correlated phenotypes in genome-wide association studies. The American Journal of Human Genetics, 94(5), 662-676.

3) For a resources paper, it is especially important to clarify the way the data and accessions are made available to the community. To ensure accessibility of the accessions, seeds should ideally be deposited in a genebank, where they are propagated and can be ordered online. Otherwise, it might be difficult for others to get hold of the seeds in practice. If that is not possible, please specifically state that the seeds can be obtained from the authors. The genomic data should ideally be made available through a public genome browser, or alternatively deposited in appropriate databases (including the vcf files).

Most of the accessions are available from the IPK Gatersleben and the USDA genebanks. Materials that are not available from the genebanks can be obtained from the authors with a Standard Material Transfer Agreement (SMTA). Genomic data (Ready to use vcf files) and phenotypic data are made available through the Dryad repository https://doi.org/10.5061/dryad.zgmsbcc9m. Raw sequencing data are available from NCBI SRA. Also, detailed descriptions of the germplasm, phenotyping methods, and phenotypes are posted at https://quinoa.kaust.edu.sa/#/ and published in Stanschewski et al., 2021. Custom scripts used for creating plots are provided at https://github.com/DilanSarange/quinoaDPgwas. (see lines 584-588 and 591-592)

4) There was also a consensus that the description of the candidate genes was too assertive, as the associations do not demonstrate causality at this stage. This should be toned down.

Thank you for this comment. We modified the text accordingly (Lines 246-247, 249, 252, 264, 265, 266, 272-273, 276-277, 281, 288-289, 373-377, 378, 383, 384, 385, 391-393, 396-397, 400-404,407-409).

5) Regarding SNP calling, reviewer 3 made important suggestions to evaluate robustness of the results to a number of potential pitfalls (coverage and repetitive regions):– The correlation between coverage and heterozygosity levels represents a serious issue that should be addressed. It seems that you need more than 6x coverage to achieve accurate calling. The obvious solution would be to carry out additional sequencing, but this would cause significant delays. Another option would be to scrutinize the information in the VCF files. Many SNP-callers, including GATK, tend to err on the side of caution and call heterozygotes rather than homozygotes in difficult cases. You may well be able to salvage accurate data from the low-coverage individuals by adjusting genotype calls based on the genotype likelihoods. You could require compelling evidence for a heterozygote before it is called, instead favoring the most like homozygous genotype. Please provide a scatter plot of depth versus heterozygosity level to demonstrate that the issue was resolved.

We addressed your concern by providing the scatter plot as requested. We also calculated correlations between coverage and heterozygosity (Figure 1—figure supplement 2b). However, correlations were not significant, and therefore we believe that the coverage was sufficient enough to achieve accurate SNP-calling (lines 107-109).

– Although you have very reasonably applied a mapping quality filter to reduce the problem of poor-quality SNPs derived from repetitive regions, problems could persist. If I read Table S5 correctly, there is an overwhelming majority of intergenic SNPs. It would be interesting to see what happens if you eliminate SNPs in repetitive regions, as determined by RepeatMasker or similar, and then rerun the GWAS. If the repetitive regions contribute false positive SNPs with random genotype calls, you should get cleaner results with more significant associations.

Thank you for this suggestion; we understand the problem could occur due to the poor/incorrect mapping in the intergenic regions. Therefore, we applied stringent filtering to remove SNPs with more than 50% missing genotype data, minimum mean depth less than five, and minor allele frequency less than 5% for the GWAS analysis. SNP densities in intergenic regions are generally higher than in the genic regions. In the Supplementary file 1e, there are 511 (47% of all association) intergenic SNPs and 300 upstream or downstream (28%) that are associated with traits. Therefore, we do not think that we have an overwhelming majority of intergenic SNPs. Also, we believe that SNPs within repetitive regions are also important. For instance, repetitive elements can have a function in controlling gene expression. Moreover, since our SNP calling and filtering criteria were very stringent, the probability of having false positives in our SNP data set is very low. Therefore, we would not remove them from the GWAS analysis at this stage.

– A specific example is lines 218-227, where you argue for mapping resolution. The argument does not appear strong as none of the top SNPs in the region are located in the two candidate genes according to Figure S14b. Also it is unclear from that figure how many other genes reside within the region. It would be preferable to use the extent of LD compared to gene density as a general argument for mapping resolution.

Thank you for this comment. This is true that none of the top SNPs associated with saponin content are located in the candidate genes. However, based on local LD analysis, these genes are in LD with the highly associated SNPs. We identified in total 29 genes in this region of chromosome 5B. Two *BBLH25* genes are in LD with the significantly associated SNPs. We updated the figure to make it easy to read by excluding the non-significant SNPs from the LD heatmap except SNPs on the genes (Figure 4—figure supplement 1). Modified line 252-253, 378

[Editors’ note: further revisions were suggested prior to acceptance, as described below.]

The manuscript has been improved but there are some remaining issues that need to be addressed, as outlined below:1 – The section on « genomic patterns of variations between highland and lowland quinoa » has been expanded, but unfortunately at this stage it remains too descriptive. We understand from your response letter that a more comprehensive analysis of the demographic and selective history of quinoa is in preparation, so we suggest that this section could be shortened to include only : (1) the strong population differentiation between highland and lowland accessions, (2) the comparison of LD decay, nucleotide diversity and Tajima's D between highland and lowland, (3) the local PCA and a comparison of average FST across chromosomes to test for heterogeneity of patterns of differentiation. e.g. Do some chromosomes show a lower FST and different contribution to the three corners of the PCA, in particular Cq6B, as would be expected if it experienced recent introgression?

The information required by the reviewer is presented in Figure 1—figure supplement 4, Figure 1—figure supplement 7, Figure 1—figure supplement 8, and Figure 1—figure supplement 9. However, as the reviewer suggested, we prefer to keep this section short and leave the more in depth analysis of the population differentiation for the follow up study of sequence analysis of a diversity panel of 900 different quinoa accessions that is currently under preparation, as this diversity set provides an extensive level of genetic diversity that can be exploited for studying the breeding history in quinoa. Therefore, we believe that the point with chromosome Cq6B is beyond the scope of our analysis and could be extensively addressed in the follow-up study.

2 – The issue raised by reviewer 3 that the modest sequencing depth could lead to inaccurate SNP calling was correctly addressed by plotting mean heterozygosity vs sequencing depth across the 310+ accessions, but it is incomplete. The reviewer suggested that the genotype likelihood threshold to call heterozygous sites could be adjusted for each accession by examining the mean sequencing depth and mean genotype likelihood between heterozygous and homozygous sites. The eventual difference of means between accessions with high vs low overall coverage could be used to adjust the threshold.

We thank the reviewer for their comment. To address the reviewer’s concern, we provided Figure 1—figure supplement 3. In this figure we provided the genotype quality (GQ) of samples with high and low read depth at both homozygous and heterozygous SNP loci. Here we show that there is no significant differences between GQ resulting from low sequencing depth, affecting particularly heterozygous sites. Combining these data with the fact that there is also no correlation between sample depth and heterozygosity level (Figure 1—figure supplement 2b), we can conclude that the modest sequencing coverage does not particularly affect the quality of heterozygous SNPs calling.

3 – Along the same lines, a missing piece of basic information is the proportion of SNPs called as homozygous vs heterozygous. Given the high selfing rate, a high proportion of homozygous SNPs is expected. Is this observed, and does that vary across accessions?

Supplementary file 1b provides this information for every accession. As expected, the number of homozygous SNPs are much higher compared to heterozygous SNPs in all quinoa accessions.

4 – Finally, reviewer 3 had also suggested restricting the set of SNPs to those outside repeated regions, as identified by e.g. RepeatMasker in the reference genome. Even using stringent filters to identify SNPs will not totally alleviate the problem of SNP calling in repeated regions, and these will remain dubious. Please do consider testing whether the marker-trait associations detected are still detected (and perhaps more convincingly so) when removing SNPs in repeated regions.

In response to the reviewer’s concern, we compared the number of MTAs between whole genome SNP set and repeat masked SNP set. This resulted in the removal of all repetitive sequences accounting for 64% of the quinoa genome, including retrotransposons and other sequences which are known to be dispersed also across gene-rich regions. Therefore, the GWAS resulted in much less markertrait associations. Only, 27% of the MTAs were in common between the two datasets (see Figure 3—figure supplement 7a). However, the comparison of GWAS results between whole genome and repeat masked SNP set showed that highly significant associations could be identified even if the repetitive regions were excluded from the analysis (Figure 3—figure supplement 7b and c). This comparison yielded similar results for single traits associations across both years. It is likely that the associated SNPs with traits identified after removing the repetitive sequences are not the exact same SNPs when considering the whole genome data, but they are in linkage disequilibrium with them. Hence, the same regions of the genome would still show significant associations in GWAS analysis, even after removing repetitive sequences, which was perfectly demonstrated in case of the pleiotropic locus on chromosome Cq2A (Figure 3—figure supplement 7c). Moreover, we believe that completely removing repetitive sequences is not a suitable strategy.

Furthermore, we only considered SNPs that showed consistent associations with traits in both years as MTAs. Therefore, while we do not claim to find all the MTAs in the quinoa genome, we believe we have identified the most consistent MTAs based on our experimental and environmental conditions.

Additionally, we haven’t found any recent association study (a few listed below), where SNPs in repetitive regions were removed before GWAS.

Bheemanahalli, R., Knight, M., Quinones, C., Doherty, C. J., and Jagadish, S. V. (2021). Genome-wide association study and gene network analyses reveal potential candidate genes for high night temperature tolerance in rice. Scientific Reports, 11(1), 1-17.

Kang, L., Qian, L., Zheng, M., Chen, L., Chen, H., Yang, L., … and Liu, Z. (2021). Genomic insights into the origin, domestication and diversification of Brassica juncea. Nature genetics, 53(9), 1392-1402.

Mwando, E., Han, Y., Angessa, T. T., Zhou, G., Hill, C. B., Zhang, X. Q., and Li, C. (2020). Genome-wide association study of salinity tolerance during germination in barley (Hordeum vulgare L.). Frontiers in plant science, 11, 118.

Pang Y., Liu C., Wang D., St. Amand P., Bernardo A., Li W., He F., Li L., Wang L., Yuan X., Dong L., Su Y., Zhang H., Zhao M., Liang Y., Jia H., Shen X., Lu Y., Jiang H., Wu Y., Li A., Wang H., Kong L., Bai G., and Liu S. (2020). High-Resolution Genome-wide Association Study Identifies Genomic Regions and Candidate Genes for Important Agronomic Traits in Wheat. Mol. Plant. 13, 1311–1327.

Varshney, R. K., Roorkiwal, M., Sun, S., Bajaj, P., Chitikineni, A., Thudi, M., … and Liu, X. (2021). A chickpea genetic variation map based on the sequencing of 3,366 genomes. Nature, 1-6.

Zia, M. A. B., Demirel, U., Nadeem, M. A., and Çaliskan, M. E. (2020). Genome-wide association study identifies various loci underlying agronomic and morphological traits in diversified potato panel. Physiology and Molecular Biology of Plants, 26(5), 1003-1020.

5 – The text contains too many acronyms that are rarely useful. Examples of acronyms that are not repeated frequently and could make the text a lot easier to follow if the words were instead written in full include : HCSNPs, NoB, STL, DTM, PL, PH, TSW, GT, DTB, DTF, PCA(PHEN), BLUE, MTA, BBCH60, BBCH94.

The abbreviations are now mentioned in full term as suggested by the reviewer. However, we would still prefer to leave the acronyms PCA(PHEN), MTA, BBCH60 and BBCH94 as they represent a either a specific type of analysis or a specific scale for the phenological development stage of the plants that cannot be described in full terms every time it is mentioned in the text.